# Providing New Insights on the Molecular Properties and Thermal Stability of Ovotransferrin and Lactoferrin

**DOI:** 10.3390/foods12030532

**Published:** 2023-01-25

**Authors:** Qi Zeng, Yaping Liu, Jing Sun, Yongguo Jin

**Affiliations:** 1College of Food Science and Technology, Huazhong Agricultural University, Wuhan 430070, China; 2College of Food Science and Engineering, Northwest A&F University, Xianyang 712100, China; 3Institute of Animal Husbandry and Veterinary, Hubei Academy of Agricultural Science, Wuhan 430072, China

**Keywords:** Ovotransferrin, lactoferrin, pH effect, thermal stability, protein structure

## Abstract

Ovotransferrin (OVT) is a multi-functional protein showing over 50% homology with Bovine lactoferrin (BLF) and human lactoferrin (HLF), which have the potential to be a substitute for lactoferrin (LF) due to the limited production of LF. To explore the substitutability of OVT, the molecular properties and thermal stability of OVT, BLF and HLF were characterized because these properties will affect the processing quality and biological activities of protein products when exposed to different processing conditions (e.g., temperature, pH, ion strength). The results showed that although obviously different isoelectric point (5.31, 9.12 and 8.75 for OVT, BLF and HLF, respectively), particle size distribution and hydrophobicity were found, they exhibited good dispersity because of high potential value. They showed an endothermic peak at 80.64 °C, 65.71 °C and 90.01 °C, respectively, and the denaturation temperature varied at different pH and ionic strength. OVT and BLF were more susceptible to heating at pH 5.0 as reflected by the decline of denaturation temperature (21.78 °C shift for OVT and 5.81 °C shift for BLF), while HLF could remain stable. Compared with BLF, OVT showed higher secondary structure stability at pH 7.0 and 9.0 with heating. For example, the α-helix content of OVT changed from 20.35% to 15.4% at pH 7.0 after heating, while that of BLF changed from 20.05% to 6.65%. The increase on fluorescence intensity and redshifts on the maximum wavelength after heating indicated the changes of tertiary structure of them. The turbidity measurements showed that the thermal aggregation degree of OVT was lower than BLF and HLF at pH 7.0 (30.98%, 59.53% and 35.66%, respectively) and pH 9.0 (4.83%, 12.80% and 39.87%, respectively). This work demonstrated the similar molecular properties and comparable thermal stability of OVT to BLF and HLF, which can offer a useful reference for the substitute of LF by OVT.

## 1. Introduction

Bovine lactoferrin (BLF) and human lactoferrin (HLF) constitute of a polypeptide chain with 689 and 691 amino acid residues, separately. They are the members of the transferrin family and have identical iron-binding sites. They all possess two homologous symmetrically globular lobes (N-terminal and C-terminal lobe), each type of lobe can be divided into N1 and N2, C1 and C2, respectively, which give them the capacity to bind ferric ions [1,2]. Apart from iron-binding activity, some in-vitro, in-vivo and clinical studies have reported that lactoferrin (LF) can show antioxidation [3], anticancer [4], anti-inflammatory [5], immunomodulating [6] and antimicrobial activities [7]. There are 0.02–0.35 mg/mL LF in bovine milk and 1.0–3.2 mg/mL LF in human milk [8,9,10]. LF is often extracted from bovine milk and applied in the fields of food, medicine, cosmetics etc. For example, it has been added into the infant formula to improve the immunity of infants, and used as an important ingredient in special medical food and health-care food to assist in the treatment of diseases and enhance human health [11]. In China, due to the weakness of large-scale preparation technology on LF, the commercial LF is mainly dependent on importing from other countries, such as Netherlands, New Zealand, Australia and America. The national food standard (GB1903.17-2016) for food nutrition fortifier has increased the purity requirement of LF to 95%, and it is hard for most of the LF manufacturers to reach the requirement. As a result, the import volume of qualified LF product has been decreasing and thus the price of LF has been rapidly rising due to the limited supply [12]. Under this circumstance, finding a similar functional protein to supplement the deficiency of LF would be an alternative. Ovotransferrin (OVT), the high abundance protein (about 12% of total proteins) in the egg white, is a single-chain glycoprotein consisting of 686 amino acid residues [13]. It shows around 52% homology to LF in terms of amino acid sequence and they have similar N-terminal and C-terminal lobes. Additionally, some studies have shown that OVT has various biological activities consistent to LF, such as antimicrobial [14], anti-inflammation [15], immunomodulation [16], anti-oxidation [17] and bone-promoting activity [18]. Since the sources of egg are more abundant than these of milk and the purification technology of OVT is relatively mature, OVT is expected to be a potential substitute for LF [19,20,21,22]. Following this, the application of OVT in the field of functional foods would be extended and the insufficiency of LF yield can be addressed.

Processing properties are important for proteins as a functional ingredient in commercial use, which are generally relevant to the structure, isoelectric point, dispersibility, thermal denaturation and aggregation behavior of proteins. Thermal denaturation exhibits a dominating hurdle in the wider application of protein in food products as unique functional ingredients. Ionic strength and pH are demonstrated to play a significant role in the heat-induced unfolding and aggregation of protein molecules [23]. The research results of Liu et al. showed that ovomucin-depleted egg white proteins were most sensitive to heat-induced protein aggregation at pH 5.0 [24]. Ionic strength could influence the thermal stability of protein in aqueous solution by changing the hydration behavior and electrostatic interaction of protein molecules. Heat denaturation is also highly relevant to the conformation changes on the advanced structures of protein. Driven by heating, protein may experience some disruptive processes, e.g., the misfolding and unfolding of structure, the secondary structure of protein tends to transform from the ordered into the unordered structure [25,26]. It has been reported that OVT and LF can show various functional properties that can be affected by the different degrees of protein denaturation because the biological activities of protein mainly rely on the correct folding of native state. Protein thermal denaturation commonly occurs during food processing (e.g., pasteurization) [27]. Therefore, better thermal stability is vital to retain the desirable biological activities of functional proteins. As for the feasibility of OVT substituting for LF, OVT is expected to have comparable or even better thermal stability to LF. In this case, OVT will have the advantage to retain its natural biological activities and show better product processing quality. Under this circumstance, the similarities and differences on the molecular properties and thermal stability between OVT and LF need to be explored and characterized. In this work, SDS-PAGE, dynamic light scattering, fluorescence spectroscopy, X-ray diffraction, nano differential scanning calorimetry (Nano DSC), circular dichroism (CD) and UV-visible spectroscopy were used to investigate the molecular weight distribution, sulfydryl content, isoelectric point, particle size, hydrophobicity, crystalline state, structure stability, thermal denaturation and aggregation behavior of OVT, BLF and HLF.

## 2. Material and Method

### 2.1. Materials and Reagents

SDS-PAGE kit was purchased from Beijing Labgic Science&Technology Co., Ltd. (Beijing, China). Human lactoferrin (HLF, holo-form, purity ≥ 95%) and bovine lactoferrin (BLF, holo-form, purity ≥ 95%) was purchased from Sigma-Aldrich Co., (Saint Louis, MI, USA) and Shanghai yuanye Bio-Technology Co., Ltd. (Shanghai, China), respectively. Ovotransferrin (OVT, holo-form, purity > 95%) was prepared using Fe^3+^ saturation and two-step ethanol precipitation method. 5,5′-dithiobis(2-nitrobenzoic acid) (DTNB) and 8-anilino-1-naphthalenesulfonic acid (ANS) were purchased from Shanghai Macklin Biochemical Technology Co., Ltd. (Shanghai, China). Trichloroacetic acid, tris base, glycine and other chemicals of analytical grade were purchased from Sinopharm Co., (Beijing, China). Additionally, 2 × SDS loading buffer (100 mM pH 6.8 Tris buffer, 200 mM DTT, 4% SDS, 20% glycerin and 0.1% bromophenol blue) was purchased from Shanghai Beyotime Biotechnology Co., (Shanghai, China).

### 2.2. Preparation of Ovotransferrin

Ovotransferrin was prepared using Fe^3+^ saturation and two-step ethanol precipitation method reported by Abeyrathne et al. [28]. Briefly, the egg whites were diluted with equal volume of pure water, followed by stirring at 4 °C for 2 h. The egg white solution was then mixed with 20 mM FeCl_3_ · 6H_2_O solution at a volume ratio of 40:1. Subsequently, ethanol solution with a final concentration of 43% (*v*/*v*) was added to the obtained egg white solution, and the mixture was centrifuged at 5000 r/min and 4 °C for 20 min. The ethanol concentration of the obtained supernatant was further adjusted to 59% (*v*/*v*), followed by centrifugation at 4 °C. The collected precipitates were re-dissolved with 43% ethanol solution and then treated with 59% (*v*/*v*) ethanol. The resultant precipitates were re-dissolved with 9 times volume of pure water and the solution was concentrated by rotary evaporation to remove ethanol. Finally, the concentrated sample was freeze-dried in vacuum and stored at −20 °C for further use.

### 2.3. Molecular Weight Distribution

OVT, BLF and HLF solution (2 mg/mL) were separately mixed with 2 × SDS loading buffer (non-reducing and reducing) at a volume ratio of 4:1, followed by boiling water bath for 5 min. The separating gel and stacking gel with the concentration of 12% and 5% were prepared using a SDS-PAGE kit, respectively. Samples and protein marker (10–180 kDa) with 10 μL were loaded onto each lane and the gel electrophoresis experiment was conducted using an electrophoresis system (DYY-10C, Beijing Liuyi Co., Ltd., Beijing, China) with stacking gel voltage of 80 V and separating gel voltage of 120 V. After the electrophoresis was finished, the gel was fixed, stained and destained in sequence [29].

### 2.4. Determination of Sulfhydryl Content

For free sulfhydryl, 0.2 mL of 5 mg/mL OVT, BLF and HLF solution was mixed with 1 mL of 8 M urea-Tris-Gly solution (pH 8.0), followed by adding 20 μL 4 mg/mL Ellman’s reagent. After the mixture was reacted at room temperature for 30 min, the absorbance of solution was measured at 412 nm. For total sulfhydryl, 0.2 mL of 5 mg/mL OVT, BLF and HLF solution were mixed with 1 mL 10 M urea and 20 μL β-mercaptoethanol, followed by reacting at room temperature for 1 h. After that, the obtained solutions were treated with 5 mL 12% TCA for 1 h and centrifuged at 5000× *g* for 10 min. The precipitation and centrifugation steps were repeated two times. Next, the precipitates were redissolved in 2 mL 8 M urea-Tris-Gly solution (pH 8.0), and 20 μL Ellman’s reagent was added. The absorbance of the solution was finaly measured at 412 nm [30]. The sulfhydryl content was caculated by the following Equation (1):*SH* (μmol/g protein) = 75.53 × *A*_412_ × *D*/*C*(1)
where *A*_412_ is the absorbance of mixture at 412 nm, *D* is the dilution factor, *C* is the concentration of OVT, BLF and HLF

### 2.5. Determination of Particle Size and Potential

The average diameter (D_Z_), polydispersity index (PDI) and zeta potential of protein solution (1.0 mg/mL) were determined using a dynamic light scatting instrument (Zetasizer Nano ZS, Malvern Instrument Co., Ltd., Worcestershire, UK) according to a previous method [31]. D_Z_ and PDI were determined at pH 7.0. The isoelectric point of three proteins was obtained by measuring the potential of protein solution at the pH range of 4.5–6.5 for OVT, 6.0–10.5 for BLF an HLF.

### 2.6. Measurements of Surface Hydrophobicity

The protein solutions with concentrations of 0.025, 0.05, 0.10, 0.20, 0.40, 0.60 and 0.8 mg/mL were prepared using phosphate buffer (0.01 M, pH 7.0). ANS (8 mmol/L, 15 μL) was added into 2 mL protein solution, followed by shaking and incubation at room temperature for 1 h. The fluorescence intensity of the mixtures was then measured at the excitation wavelength of 390 nm (slit 5 nm) and the emission wavelength of 470 nm (slit 5 nm) using a fluorescence spectrophotometer (RF-5301pc, Hitachi co., Tokyo, Japan). The fluorescence intensity was plotted against the protein concentration to obtain a standard curve, and the slope of the curve was expressed as the surface hydrophobicity index (H_o_). The visualized 3D structures of OVT, BLF and HLF were drawn using VMD (Visual Molecular Dynamics) 1.9.4 software (Urbana-Champagne, IL, USA).

### 2.7. X-ray Diffraction (XRD)

The XRD patterns of protein powders were recorded by X-ray Diffractometer (D8 ADVNCE, Malvern Instruments Ltd., Malvern, Britain). Measurements were carried out at an angle range of 5–80° with a scan rate of 2°/min, divergence slit of 1° and receiving slit of 0.1 mm for the incident beam. The patterns obtained were analyzed by Jade 6.0 (MDI Co., Livermore, CA, USA).

### 2.8. Nano Differential Scanning Calorimetry (Nano DSC)

Protein solutions with concentrations of 1 mg/mL, pH of 5.0, 7.0 and 9.0, and ionic strengths (NaCl) of 0.145 M, 0.290 M and 0.580 M were prepared and degassed for 10 min by a degassing station referring to a previous study [32]. Two proteins with 1 mg/mL were mixed at equal volume. Measurements were conducted using Nano differential scanning calorimeter (TA Instruments Co., New Castle, PA, USA) with the temperature range of 40–100 °C, heating/cooling rate of 1 °C/min and constant pressure. The ultrapure water and protein solution (about 500 μL) were injected into the reference and sample cell using a micropipetter, respectively. After the measurements were ended, the Nanoanalyze^TM^ software was used to analyze the raw data.

### 2.9. Circular Dichroism

The circular dichroism (CD) spectra of three protein solutions (0.5 mg/mL) before and after heated at 70 °C for 30 min with pH 5.0, 7.0 and 9.0 were recorded using a J-810 CD spectrometer (JASCO Co., Ltd., Tokyo, Japan) with wavelength range of 190–240 nm, light path length of 0.1 cm, scanning speed of 100 nm/min and resolution of 20 mdeg. The secondary structure content was calculated using Spectra Manager software following the Yang’s algorithm [33].

### 2.10. Endogenous Fluorescence Spectroscopy

Fluorescence spectra were scanned using a fluorescence spectrophotometer (RF-5300, Shimadzu Co., Kyoto, Japan), where the excitation wavelengths were set at 295 nm, and the emission wavelength range was set at 300–450 nm. The excitation slit width and emission slit width were both 10 nm [34]. The changes of fluorescence intensity of three protein solutions (0.5 mg/mL) before and after heated at 70 °C for 30 min with pH 5.0, 7.0 and 9.0 were recorded. 

### 2.11. Determination of Turbidity

The absorbance values of three protein solutions (0.5 mg/mL) before and after heated at 70 °C for 30 min with pH 5.0, 7.0 and 9.0 were recorded at 400 nm using a spectrophotometer (NANO 2000C, Thermo Fisher Scientific Co., Ltd., Waltham, MA, USA). According to a previous study [35], the absorbance (A) was transformed into transmittance (T) based on the equation (A = −lgT), and the turbidity (%) was reported as 100 − T (%).

### 2.12. Statistical Analysis

The experimental data were expressed as means ± standard deviation for triplicates. One-way analysis of variance (ANOVA) and Duncan’s multiple-range tests were taken to analyze the significant differences of the data using IBM SPSS Statistics 22.0 software (IBM Corporation, Armonk, NY, USA). The results were considered statistically significant at *p* < 0.05.

## 3. Results and Discussion

### 3.1. Characterization of Molecular Properties

#### 3.1.1. Molecular Weight Distribution

Ovotransferrin (OVT), bovine lactoferrin (BLF) and human lactoferrin (HLF) comprise of 686, 689 and 691 amino acids, respectively, and they all have a molecular weight of around 78 kDa that varies based on the post-translational modification (e.g., glycosylation, phosphorylation) [30]. As depicted in Figure 1, with non-reducing and reducing electrophoresis, the three transferrins all showed single band around 80 kDa, no extra band appeared. There was no difference on molecular weight of these three proteins between non-reducing and reducing group because they only possess a single polypeptide chain that is folded into two symmetrical globular lobes (the N-terminal and C-terminal lobes) with the assistance of disulfide bonds and other intermolecular forces [36]. When the reducing reagent (β-mercaptoethanol) broke the disulfide bonds of protein, the single polypeptide chain was still intact and thus the molecular weight did not change.

#### 3.1.2. Determination of Sulfhydryl Content

Sulfhydryl groups of cysteine can be dehydrogenated to form disulfide bonds that are an important covalent bond force for the formation of advanced structure of protein. As can be seen from Table 1, free sulfhydryls were not detected in OVT, BLF and HLF, which means that all the free sulfhydryls from cysteine residues have formed intramolecular disulfide bonds [37]. Total sulfhydryl content can reflect the number of disulfide bond which is related to the structure stability of protein in different processing conditions (e.g., pH, temperature, ionic strength). Protein molecule with more disulfide bonds generally tends to have higher thermostability [38]. The content of total sulfhydryl for BLF (202.01 μmol/g) and HLF (204.32 μmol/g) was nearly equal, which is significantly higher than that of OVT (196.10 μmol/g) (*p* < 0.05). Actually, OVT contains 30 cysteines with 15 disulfide bonds that are positioned on Cys-Cys residues [39]. There are 17 disulfide bonds (34 cysteines) for BLF on Cys-Cys residues [40]. HLF with 32 cysteines contains 16 disulfide bonds on Cys-Cys residues [41]. Some research on three-dimensional structure of transferrins by X-ray crystallography have reported that these three transferrins all have six disulfide bridges in the N-terminal lobe, including two in the N1 domain, three in the kringle of domain 2, and one in the N2 domain, but no interdomain disulfide bridges exist in the N-terminal lobe. There is also a six-disulfide bridge motif conserved in the C-terminal lobe. Additionally, the extra disulfide bridges are existed in the C-terminal lobe, which are relevant to the asymmetric structure and functional characteristics of the two lobes of the transferrins [41,42]. The results suggested that the number and position of disulfide bonds of OVT, BLF and HLF were generally different, but they had similar content of total sulfhydryl.

#### 3.1.3. Determination of Particle Size and Potential

OVT is an acidic glycoprotein comprising of 101 positively charged and 91 negatively charged amino acid residues, while BLF and HLF is a basic glycoprotein with 101, 98 positively charged amino acid residues, and 76, 79 negatively charged amino acid residues, respectively [43,44]. The isoelectric point (pI) of a single protein is the pH at which the potential of protein solution is zero. It is clear from Figure 2A that the pI of OVT was 5.31, which was obviously lower than that of BLF (9.12) and HLF (8.75). This difference was attributed to the fact that OVT have more negatively charged amino acid residues (Asp and Glu) than BLF and HLF [41]. Additionally, the chemical modification (such as phosphorylation) of protein might cause the shift of pI [45]. In the solution system with neutral pH, OVT are negatively charged, BLF and HLF are positively charged, all with high absolute potential value. This means that, in spite of the difference on pI among them, they can maintain stable in the solution with neutral pH during practical processing because of the high potential value [46].

Dynamic light scattering (DLS) is a useful tool to characterize the protein size distribution and aggregation. Different sizes of protein particles in the solution can contribute to distinctive fluctuations of scattered light [47]. Figure 2B shows that OVT possessed an average particle size of 429 nm with PDI of 0.570 at a neutral pH, whereas BLF and HLF had a particle size of 143 nm and 123 nm with PDI of 0.326 and 0.266, respectively. Figure 2C also exhibits the differentiated particle size distribution among these three proteins. Electrostatic and hydrophobic forces have a strong influence on the aggregation and dispersity of protein molecules [48]. The particle size of OVT might be affected by the surface charge and hydrophobicity. When the pH of the protein solution was close to OVT’s pI, the electrostatic repulsive force among OVT molecules weakened and the molecules thus tended to assemble to different size of multimers via hydrophobic interactions, which led to a larger particle size and higher PDI. Similarly, a previous study showed that egg-derived lysozyme was present as a dimer or trimer at alkaline pH near its pI [49]. As for BLF and HLF, the number of their positively charged amino acid residues is remarkably larger than their positively charged ones, which is conducive to generate strong electrostatic repulsive force for the good dispersity of protein molecules, thus contributing to a lower particle size and PDI. 

#### 3.1.4. Surface Hydrophobicity (H_0_)

Hydrophobicity is a phenomenon that the hydrophobic side chains of native proteins clump together due to their repulsion against water molecules [50]. The protein surface hydrophobicity is related to the distribution of hydrophobic amino acid residues on the surface of protein molecules, which can be reflected by the binding strength of exogenous fluorescent probe ANS to hydrophobic residues [51]. Therefore, the hydrophobic sites exposed on the protein surface can be detected to characterize the differences in protein conformation. As shown in Figure 3, BLF and HLF showed a similar hydrophobicity index (H_0_) of 910 and 955, respectively, which were significantly higher than OVT with H_0_ of 718. The hydrophobic groups of these three proteins are highlighted in blue in the visualized 3D structure. This discrepancy could be explained by the larger number of hydrophobic amino acid residues (e.g., Phe, Trp, Leu, Ala) on the primary structure of BLF and HLF. Additionally, since BLF and HLF share the same regular connecting helix based on interlobe interactions, the packing of non-polar surface on C-lobe and N-lobe forms the hydrophobic cushion [52]. Additionally, the binding of ferric iron to transferrins could affect the conformational changes (exposure of hydrophobic groups) [53]. OVT with lower H_0_ can easily hydrate in the solution and is less likely to aggregate by hydrophobic interaction, which would be advantageous to BLF and HLF in terms of processing properties in solution system.

#### 3.1.5. X-ray Diffraction Characterization

Each protein chain can be divided into an amorphous region and a crystalline region. Since the amino acid structure is rich in hydrogen bonds, driven by the hydrogen bonds, a crystalline region can form through β-sheeting, which plays a role in protein cross-linking. The amorphous chains are responsible for connecting the various crystalline regions [54]. The XRD patterns and relevant the quantitative data of OVT, BLF and HLF are presented in Figure 4 and Table 2, respectively. There were two notable broad peaks at the 2θ of around 9° and 21° in the XRD pattern of these three proteins, which were typical features for protein powder [55]. OVT possessed higher peak intensity and area than BLF and HLF, indicating that it had larger content of crystalline phases despite the existence of partial amorphous phases which were relaxed and randomly aggregated [56]. Generally, crystallization is presented as the ordered array of macromolecules via crystal packing. The two diffraction peaks indirectly belong to the ordered structure (α-helix and β-sheet) in the polypeptide chain [57]. The discrepancy of peak intensity and area could therefore reflect the difference on the secondary structures of OVT, BLF and HLF. The higher levels on α-helix and β-sheet structure of OVT than BLF and HLF were indirectly speculated, indicating the more ordered structure of OVT. The average distance between protein crystals could illustrate the difference of backbone structure, which was not significantly different among these three proteins.

### 3.2. Thermal Stability Analysis

#### 3.2.1. Nano Differential Scanning Calorimetry (DSC) Characterization

Nano DSC allows to study the thermal transition occurring in macromolecule solution at the temperature range of –10 to 130 °C and offers a new dimension in sensitivity, baseline noise and repeatability in both heating and cooling modes. It is designed to specifically measure the absolute heat capacity and denaturation temperature of macromolecules in dilute solution, thus exploring the conformation and solvation as well as subtle changes in folding of macromolecules to evaluate the thermal stability or interactions on macromolecular structure [58]. As shown in Figure 5A, 5B and 5C, OVT, BLF and HLF all presented a single sharp endothermic peak at the maximum temperature (T_m_) of 80.64 °C, 65.71 °C and 90.01 °C, respectively, indicating the highest thermal stability of HLF and the lowest thermal stability of BLF. The differences on T_m_ are related to the closeness degree of globular lobes of these three proteins [13]. The binding of ferric ion can also affect the thermal stability of transferrins. The iron-depleted (apo-form) HLF has higher susceptibility to heat treatment than iron-saturated HLF (holo-form) [59]. Besides, the glycosylation pattern are reported to influence the thermal stability of protein [60]. Thermal-aggregation is generally characterized by thermodynamic and kinetic components. Thermodynamic component causes unfolding of macromolecule structure and kinetic component could result in partial or complete irreversibility [61]. The second heating scan was conducted to determine if the unfolding of these three proteins is reversible. The diagram displays a flat curve for OVT, BLF and HLF after the second heating scan, which demonstrated that these three proteins got trapped in an irreversible denatured state. Two different proteins might interact with each other in a solution system and thus affect their thermal properties or relevant functionalities. Figure 5D shows the DSC profile of single protein, OVT-BLF and OVT-HLF mixture. When OVT was mixed with BLF, the T_m_ of both proteins slightly shifted to a higher value compared to that of single protein, suggesting that there might exist hydrophobic and electrostatic interactions between OVT and BLF during heating, which affected the unfolding state of each protein. Compared with OVT mixing with BLF, the enthalpy change (ΔH) and T_m_ of OVT decreased after mixed with HLF, indicating that the interactions between OVT and HLF weakened the thermal stability of OVT to some extent. The different pI among these three proteins shown in Figure 2A could prove the electrostatic interactions of OVT-BLF and OVT-HLF mixture at neutral pH. There are studies reporting that noncovalent bonding plays an important role in protein-protein interactions, especially in the early heating stages [62]. The higher T_m_ of OVT than BLF suggested that OVT was more thermally stable than BLF. This will be an advantage for the thermal processing of OVT.

The changes of pH can affect the surface net charge of the protein. Generally, at extreme pH, the protein molecules tend to unfold due to the exposure of buried functional groups. At intermediate pH, the ionization of side chains from the protein molecules will generate alteration of hydrogen bonding and salt bridges which could destabilize the structure of protein [63]. Figure 6 shows the pH-induced changes of the DSC thermograms of OVT, BLF and HLF. As shown in Figure 6A, compared to the pH 7.0 group, the T_m_ and ΔH of OVT solution remarkably decreased from 80.64 °C to 58.86 °C (21.78 °C shift), 572.90 to 400.30 kJ/mol at pH 5.0 and slightly increased to 81.16 °C and 692 kJ/mol at pH 9.0, respectively. As for BLF (Figure 6B), compared to the pH 7.0 group, the T_m_ and ΔH notably decreased from 65.71 °C to 59.90 °C (5.81 °C shift), 757.90 to 651.60 kJ/mol at pH 5.0 and slightly increased to 66.28 °C and 788.10 kJ/mol at pH 9.0, respectively. OVT had similar trend to BLF, but it was more sensitive to heat treatment at pH 5.0 than BLF. The significant decrease in T_m_ at acidic condition suggested that the advanced conformation of OVT and BLF obviously changed, which in turn affected the rate of protein unfolding and aggregation during heating. At acidic condition, hydrogen ions tend to react with amino groups on the protein structure to generate –NH_3_^+^ groups, which will change the intermolecular or intramolecular hydrogen bonding and electrostatic forces, and thus the advanced structure of protein molecules [64]. Additionally, acidic condition could weaken the iron-binding ability of these two proteins by changing the movements of side chains and lobes, and promote ferric ion release, thus reducing the thermal stability [65]. Sreedhara et al. reported that when the pH of protein solution was gradually adjusted from 7.0 to 3.0, a continuous decrease on the T_m_ of BLF was observed and the T_m_ reached the lowest value of 39 °C [1]. At alkaline condition, OVT and BLF could reveal a higher thermal stability based on electrostatic repulsion force [66]. A previous study reported that pH altered the T_m_ of the pea proteins and higher pH values contributed to higher T_m_ and ΔH [67]. HLF exhibits the highest T_m_ (90.01 °C) and ΔH (999.40 kJ/mol) at pH 7.0 (Figure 6C), and the T_m_ and ΔH slightly shifted to 88.28 °C, 953.40 kJ/mol at pH 5.0, and to 89.37 °C, 977.60 kJ/mol at pH 9.0, respectively. It generally maintained a good thermal stability at different pH despite a slight shift of T_m_ and ΔH, which could be explained by the high flexibility (including relative movements of C-lobe and N-lobe) of tertiary structure of HLF [68].

Ionic strength can influence the molecular properties of proteins. Low ionic strength tends to promote the hydration of protein molecules, but high ionic strength can attract a large number of water molecules to hydrate with these salt ions, which will lead to the exposure of hydrophobic group to the surface of protein structure [69]. In this case, protein molecules bind to each other through hydrophobic forces and protein aggregation occurs, which will affect the rate and extent of protein denaturation during heating [70]. As can be seen from Figure 7A, there was no obvious difference on T_m_ and ΔH when OVT was present in a solution with ionic strength of 0.145 and 0.290 M, but when the ionic strength increased to 0.580 M, T_m_ and ΔH of OVT obviously shifted to a lower value (78.75°C and 401 kJ/mol). This result indicated that higher ionic strength led to lower ionic force and enhanced hydrophobic interaction, which in turn could increase the rate and extent of OVT denaturation during heating. As for BLF (Figure 7B) and HLF (Figure 7C), the three concentrations of ionic strength did not significantly change the T_m_ of BLF and HLF despite a slight rise of ΔH at ionic strength of 0.580 M, indicating that BLF and HLF were more resistant to high ionic strength. Overall, OVT and BLF were not thermally stable at acidic condition, while HLF remained thermally steady at different pH. This means that OVT is suitable to be a component in low-acid or alkaline food system during thermal processing, such as adding OVT into milk which has the pH around 6.5–6.9 [71]. The ionic strength did not have significant impact on the heat-induced denaturation of these three proteins.

#### 3.2.2. Circular Dichroism (CD) Analysis

In the secondary structure of protein, the peptide bond arrangement is highly ordered, and the directionality of the peptide bond arrangement also determines the transition and splitting of the energy level of peptide bond. Therefore, different secondary structural features of protein result in different CD band positions and absorption intensities [72]. According to the far-ultraviolet CD of the detected protein, the information of the protein secondary structure can be reflected. The CD spectra of OVT, BLF and HLF before and after heated at 70 °C with pH of 5.0, 7.0 and 9.0 were obtained. As shown in Figure 8A, the intensities of the positive band at 192 nm, two negative bands at 208 and 222 nm remarkably decreased when OVT solution was heated at different pH, and the changes at pH 5.0 were more pronounced. It meant the reduction in α-helix content of OVT after heat treatment. As for BLF (Figure 8B), with heating, the intensities of CD bands showed an obvious decrease only at pH 7.0 and 9.0, and remained stable at pH 5.0. Only slight differences were observed on the CD spectra of HLF (Figure 8C). 

In more detail, Table 3 shows the content of specific secondary structures of OVT, BLF and HLF. There were no significant changes on the secondary structures of natural OVT, BLF and HLF at different pH, suggesting that pH did not affect the protein secondary structure without heat treatment. The α-helix and β-turn content of OVT all showed a downward trend under heat treatment at different pH, while the β-sheet content increased notably, and the random coil maintained steady. These results indicated that heating could lead to the folding of OVT structure depending on the hydrophobic interactions, intramolecular hydrogen bonds, combined with van der Waals forces [73]. Interestingly, the secondary structure content of OVT generated the largest changes at pH 5.0, the α-helix and β-turn content remarkably reduced from 20.55% to 8.75%, 21.85% to 8.20%, the β-sheet content increased from 21.80% to 53.85% (*p* < 0.05). Hydrophobic interactions are suggested to play a predominant role in forming β-sheet conformations in peptide chain with a large amount of hydrophobic groups [74]. The significant increase in β-sheet content could be explained by the weaker electrostatic interactions and stronger hydrophobic interactions in OVT molecules caused by acidic condition at early heating stage, which increased the folding rate of protein. 

Heat treatment can accelerate conformational transitions of protein, leading to a faster formation of antiparallel β-sheet structure [75]. After heating, there was a pronounced decrease on α-helix and β-turn content of BLF from 20.05% to 6.65%, 19.45% to 7.80% at pH 7.0, from 21.70% to 6.75%, 21.35% to 7.90% at pH 9.0, respectively, and a substantial increase of β-sheet content from 26.35% to 61.40% at pH 7.0, from 29.20% to 54.85% at pH 9.0, was also observed (*p* < 0.05). At neutral or alkaline pH, the net charge on the side chains of BLF reduced and thus the weak electrostatic forces combined with heating changed the secondary structure, resulting in the formation of more folded conformation. However, the secondary structure content of BLF at pH 5.0 did not undergo obvious change, suggesting that acidic condition could maintain the secondary structure stability of BLF via electrostatic repulsion. The secondary structure of HLF exhibited the reduction in α-helix and β-turn content, and the increase in β-sheet content at different pH with heating, but the amount of change was lower that of OVT and BLF, indicating the good thermal and pH stability of HLF. The better thermal stability of HLF could be supported by its higher T_m_ measured by DSC at different pH and ion strength.

Surface net charge (or pI), hydrogen bonding, hydrophobicity and interactions between polar side chains have been demonstrated to play important role in the protein folding [76]. Therefore, the distinctions on the secondary structure stability of OVT, BLF and HLF at different pH might be related to them. At pH 5.0 with heating, the changes of the secondary structure content for OVT were more pronounced than those for BLF, while at pH 7.0 and 9.0 after heating, the secondary structure content for BLF changed more significantly. Generally, the natural pH of food protein solutions is close to 7.0, OVT therefore has higher stability of secondary structure than BLF during food processing.

#### 3.2.3. Endogenous Fluorescence Spectroscopy Analysis

Fluorescence spectroscopy is a highly sensitive method for studying the folding state of protein. Phenylalanine (Phe), tyrosine (Tyr) and tryptophan (Trp) have inherent fluorescent properties and can provide good fluorescent signals. Both Trp and Tyr are excited by at the wavelength of 280 nm, while at the wavelength of 295 nm, only Trp is excited. Due to their aromatic residue properties, Trp and Tyr residues are often found fully or partially buried in the hydrophobic core of protein structure [77]. The protein structure tends to unfold once the tertiary or quaternary structure is disrupted. As a result, the side chains of amino acid residues are more exposed to the hydrophilic environment of the solvent, resulting in an increase in fluorescence intensity [78].

It is clear from Figure 9A, the natural OVT solution at pH 5.0, 7.0 and 9.0 showed different fluorescence intensities at the maximum emission wavelength of 329 nm. Compared with the pH 7.0 group, the higher fluorescence intensity at pH 5.0 indicated that acidic condition caused the exposure of hydrophobic groups of natural OVT, while the lower fluorescence intensity at 9.0 revealed that alkaline condition resulted in the compactness of protein structure and some fluorescence-generated aromatic residues were concealed inside. After heating, the redshift of maximum emission wavelength occurred in the three pH groups, coupled with a pronounced increase on fluorescence intensity in pH 7.0 and 9.0 groups. This phenomenon demonstrating that heating induced the molecular unfolding and chromophore exposure of OVT to solvent, and Trp and Tyr residues thus transferred from the non-polar environment to the polar environment [79]. The natural BLF solution at pH 7.0 showed a higher fluorescence intensity at the maximum emission wavelength of 334 nm compared to the pH 5.0 and 7.0 groups which had almost equal intensity. The fluorescence intensity of BLF (Figure 9B) remarkably increased after heating at different pH with noticeable redshifts and the highest intensity was observed at pH 9.0, which indicated that heating combined with alkaline condition could make the interior structure of BLF more unfolded. Compared to the pH 7.0 group (Figure 9C), the fluorescence intensity of HLF solution at pH 5.0 and 9.0 with or without heating all experienced a slight increase at the maximum emission wavelength of around 332 nm, but no obvious redshift was observed except for a high redshift at pH 9.0 after heating. This result could be explained by the high flexibility of HLF structure that weakened the unfolding of inner conformation and stretching of peptide chain [80]. Besides, the generally higher fluorescence intensity of BLF than OVT and HLF was due to the larger number of Trp residues. In short, the significant redshifts for the maximum emission wavelength of OVT and BLF at different pH after heating occurred, indicating the changes on the tertiary structure of OVT and BLF caused by the combination of pH and heating, while the tertiary structure of HLF was slightly affected.

Overall, as for the secondary structure content of OVT, BLF and HLF measured by CD, the ordered structure (α-helix and β-sheet) content of BLF experienced the most significant changes after heating, while HLF had higher stability of secondary structure than BLF and OVT. Additionally, the fluorescence intensity of BLF showed the biggest increase after heating, while HLF had the least changes of fluorescence intensity and redshift, indicating that HLF has more stable tertiary structure. These conclusions could explain the highest T_m_ for HLF and the lowest T_m_ for BLF at different pH and ion strength, shown in Figure 5, Figure 6 and Figure 7.

#### 3.2.4. Determination of Turbidity

Turbidity is a key indicator to characterize the stability and degree of aggregation of protein solution, which can indirectly reflect the dispersion state and solubility of protein in solution [81]. It is clear from Table 4, the turbidity of natural OVT, BLF and HLF solution generally remained stable with relatively low values at different pH, indicating that no notable protein aggregation occurred among OVT, BLF and HLF. These three proteins did not aggregate even near the pI, which could be explained by the weak electrostatic interactions between proteins due to the electrostatic shielding resulted from the interference effect of ferric ions [82]. After heating at 70 °C, the turbidity of OVT solution significantly increased from 0.34% to 56.72% at pH 5.0, 1.71% to 30.98% at pH 7.0, but no obvious change occurred at pH 9.0. The changes on turbidity of OVT solution at pH 5.0 and 7.0 were attributed to the substantial unfolding of OVT conformation and some insoluble aggregates were formed via hydrophobic interactions. The higher increase in turbidity at pH 5.0 than pH 7.0 after heating might be due to the occurrence of weaker electrostatic repulsion and stronger hydrophobic interaction near the pI (5.31) of OVT [83]. A previous study showed that the heat-induced aggregation of OVT had pH dependence, the formation of aggregates were more suppressed at pH 9.0 than pH 7.0 under 65 °C [49]. The turbidity of BLF solution at different pH all showed an increasing trend, particularly a substantial rise (from 3.06% to 59.53%) of turbidity was observed at pH 7.0, demonstrating that BLF was most susceptible to aggregation at pH 7.0. The aggregation susceptibility might be relevant to the T_m_ of BLF that is under 70 °C, and BLF denatured more significant at this condition. This result is consistent with a previous report that BLF is sensitive to thermal denaturation and aggregation at neutral pH [84]. There was a consistent sharp increase on the turbidity (from around 5% to around 40%) of HLF at pH 5.0, 7.0 and 9.0, which meant that the aggregated state of HLF induced by heating was not pH-dependent.

Yang et al. [85] explored the interaction between BLF and α-lactalbumin at pH 7.0 during co-heating, and found that α-lactalbumin could cause the exposure of hydrophobic residues in the BLF structure and some new intermolecular disulfide bonds formed, which led to the formation of thermal aggregates displayed in high turbidity. The heat-induced aggregation of egg white proteins is most likely to be initiated by OVT denaturation. OVT tends to unfold into the molten globule state under heating at 55 °C and the tertiary structure is thus broken, leading to the formation of soluble aggregates [86]. Additionally, Matsudomi et al. [49] found that no aggregates were formed when OVT was heated at 65 °C at pH 9.0, which is in agreement with the turbidity of OVT at pH 9.0 in Table 4. Overall, compared to the pH conditions, heat treatment predominates over the denaturation aggregation process of OVT, BLF and HLF. OVT showed excellent thermal stability at pH 9.0 and the degree of thermal aggregation of OVT was as similar as that of HLF, but was significantly lower than that of BLF at pH 7.0. Hence, compared with BLF and HLF, less negative effects caused by heat-induced aggregation on the functional properties of OVT will occur during thermal processing.

## 4. Conclusions

To explore the feasible substitution of OVT for LF, a comparative characterization on the molecular properties and thermal stability of OVT, BLF and HLF was conducted. It is suggested that the three proteins had similar molecular weight and sulfydryl content, but obviously different pI, particle size distribution and heat denaturation temperature. It is clear from the results of circular dichroism, endogenous fluorescence spectra and turbidity that at different pH under heat treatment, the ordered secondary structure of these three proteins decreased significantly and the hydrophobic groups in the tertiary structure exposed obviously, and notable aggregation phenomena could be found. Notably, the thermal denaturation temperature, secondary structure and tertiary structure for these three proteins influenced by pH and heating showed the similar variation rule. Overall, OVT had similar molecular properties and comparable or even better thermal stability to BLF and HLF. Therefore, from this perspective, OVT has the potential to partially substitute for LF, but further investigation about the similarities and differences on the biological activities of OVT, BLF and HLF is required to prove the substitutability.

## Figures and Tables

**Figure 1 foods-12-00532-f001:**
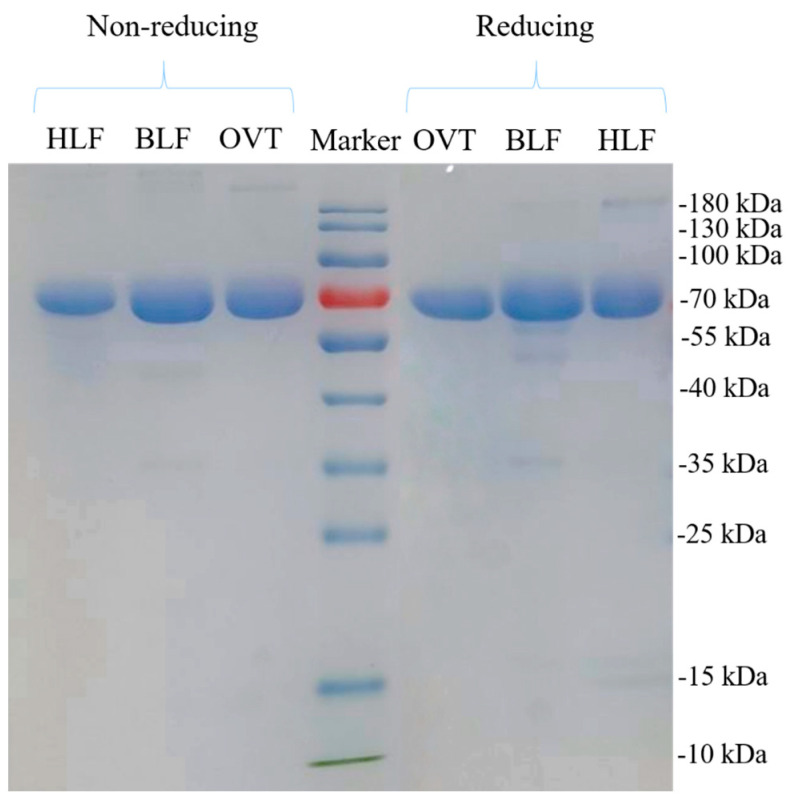
Molecular weight distribution profile of OVT, BLF and HLF. The molecular weight range of Marker is from 10 to 180 kDa.

**Figure 2 foods-12-00532-f002:**
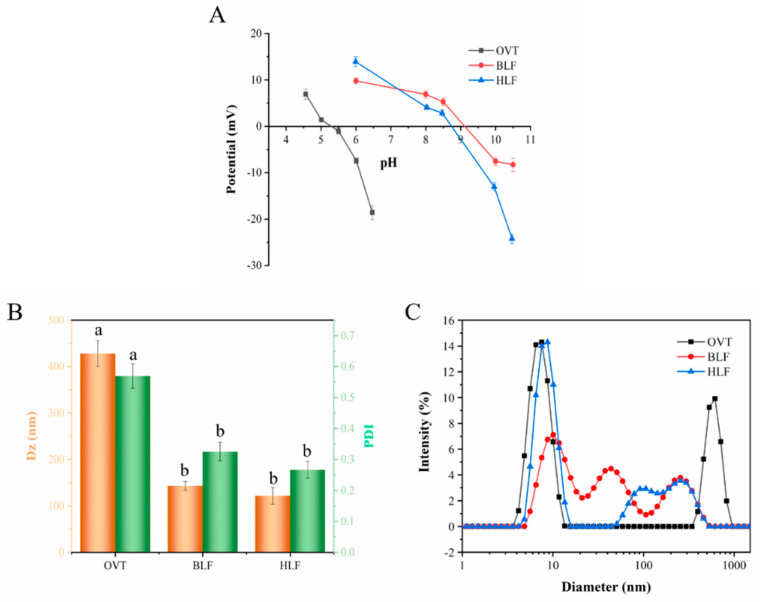
(**A**) the zeta potential of OVT, BLF and HLF solution at different pH. (**B**) The particle size and polydispersity, (**C**) size distribution of OVT, BLF and HLF solution. Different letters (a, b) indicate that the differences among the samples are significant (*p* < 0.05).

**Figure 3 foods-12-00532-f003:**
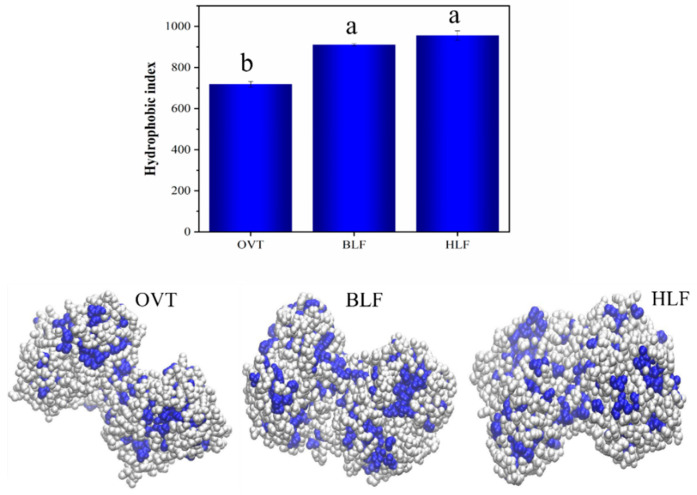
Surface hydrophobicity and visualized 3D structure (hydrophobic groups in blue) of OVT, BLF and HLF. Different letters (a, b) indicate that the differences among the samples are significant (*p* < 0.05).

**Figure 4 foods-12-00532-f004:**
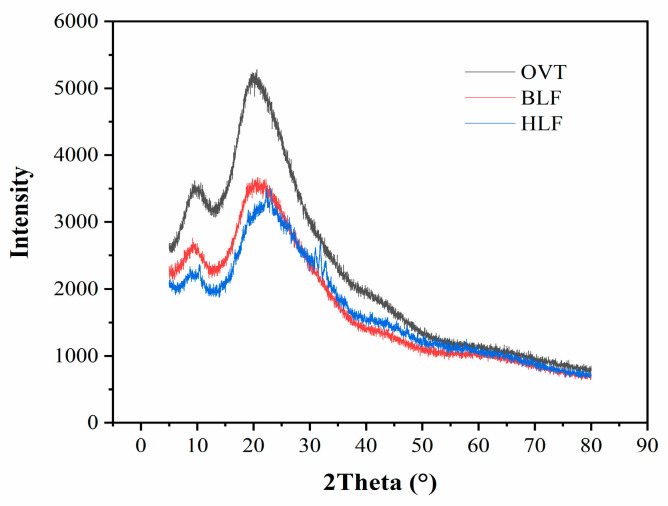
XRD patterns of OVT, BLF and HLF.

**Figure 5 foods-12-00532-f005:**
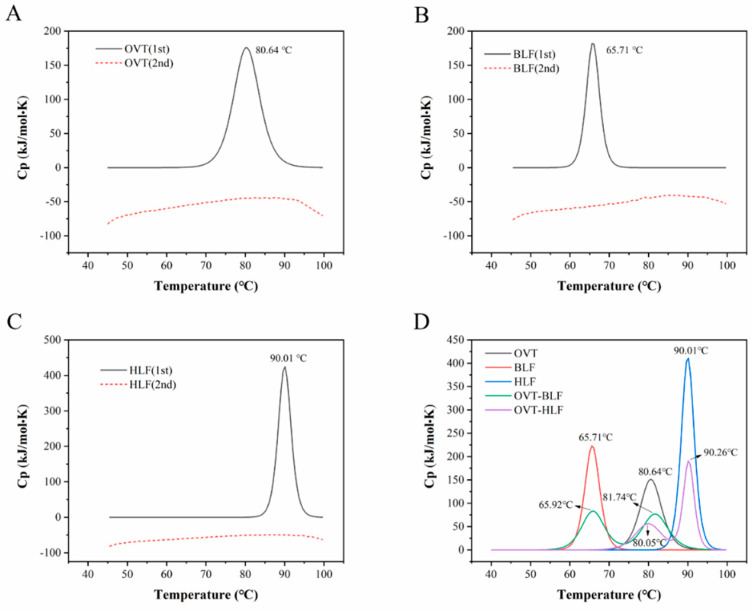
Nano DSC diagram of (**A**) OVT, (**B**) BLF, (**C**) HLF and (**D**) their mixture.

**Figure 6 foods-12-00532-f006:**
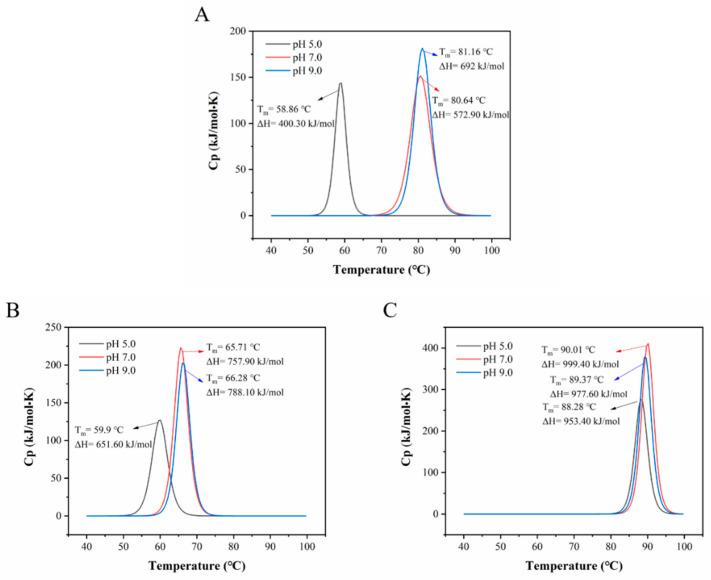
Nano DSC diagram of (**A**) OVT, (**B**) BLF and (**C**) HLF solution at pH 5.0, 7.0 and 9.0. T_m_ refers to the denaturation temperature, ΔH refers to the enthalpy change (peak area).

**Figure 7 foods-12-00532-f007:**
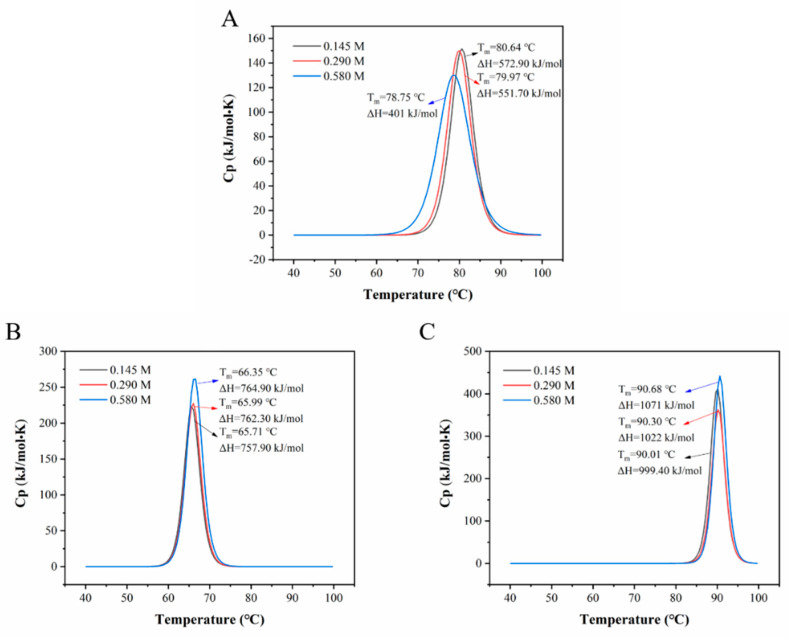
Nano DSC diagram of (**A**) OVT, (**B**) BLF and (**C**) HLF solution at ionic strength (NaCl) of 0.145, 0.290 and 0.580 M.

**Figure 8 foods-12-00532-f008:**
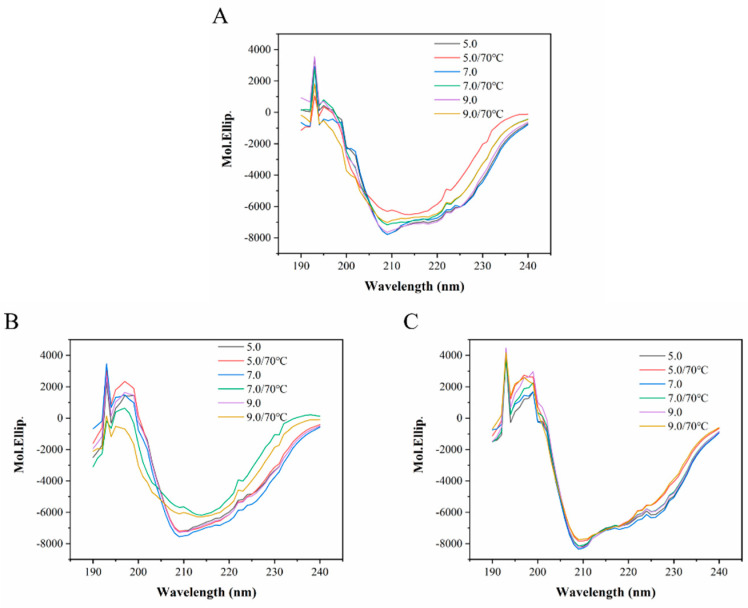
The CD spectra of (**A**) OVT, (**B**) BLF and (**C**) HLF before and after heated at 70 °C for 30 min at pH 5.0, 7.0 and 9.0.

**Figure 9 foods-12-00532-f009:**
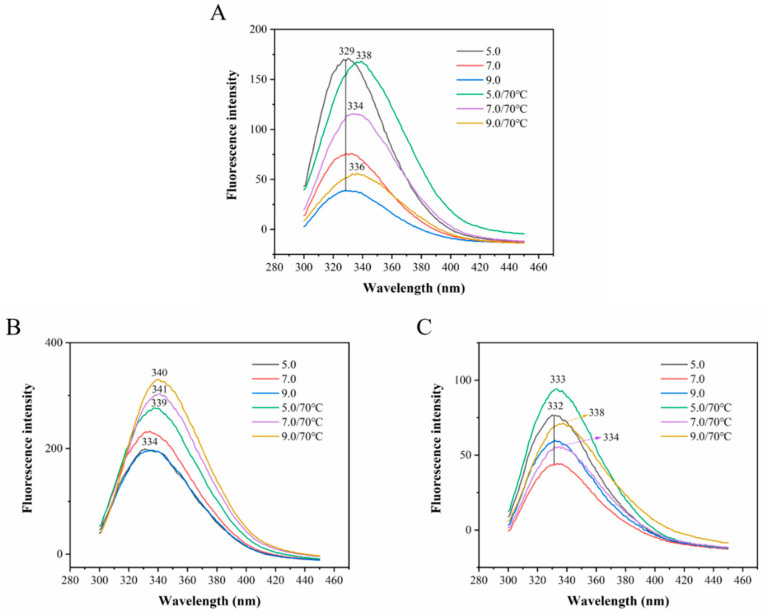
Fluorescence spectra of (**A**) OVT, (**B**) BLF and (**C**) HLF before and after heated at 70 °C for 30 min at pH 5.0, 7.0 and 9.0. Excitation wavelength: 280 nm, emission wavelength range: 300–450 nm. Each sample was measured in triplicate.

**Table 1 foods-12-00532-t001:** Sulfhydryl content of OVT, BLF and HLF.

Sample	Sulfhydryl Content	
	Free Sulfhydryl (μmol/g)	Total Sulfhydryl (μmol/g)
OVT	ND	196.10 ± 1.89 ^b^
BLF	ND	202.01 ± 3.07 ^a^
HLF	ND	204.32 ± 3.44 ^a^

Notes: Different letters (a, b) indicate that the differences among the samples are significant (*p* < 0.05).

**Table 2 foods-12-00532-t002:** XRD parameters of OVT, BLF and HLF.

Sample	2θ	Intensity	Area	Distance (Å)
OVT	9.82 ± 0.53 ^c^	449 ± 35 ^c^	95,778 ± 2631 ^d^	9.02 ± 0.16 ^b^
	21.20 ± 0.34 ^b^	1235 ± 76 ^a^	519,901 ± 5372 ^a^	4.19 ± 0.24 ^c^
BLF	8.82 ± 0.19 ^d^	202 ± 43 ^d^	32,602 ± 2132 ^e^	10.01 ± 0.25 ^a^
	21.02 ± 0.28 ^b^	816 ± 65 ^b^	346,868 ± 4627 ^c^	4.22 ± 0.33 ^c^
HLF	9.06 ± 0.22 ^d^	223 ± 28 ^d^	31,396 ± 1847 ^e^	9.76 ± 0.36 ^a^
	22.59 ± 0.55 ^a^	864 ± 63 ^b^	386,502 ± 4784 ^b^	3.93 ± 0.28 ^c^

Notes: Different letters on the same column indicate that the differences among the samples are signif-icant (*p* < 0.05).

**Table 3 foods-12-00532-t003:** The secondary structure content of OVT, BLF and HLF before and after heated at 70 °C for 30 min at pH 5.0, 7.0 and 9.0.

Sample	Treatment	α-Helix (%)	β-Sheet (%)	β-Turn (%)	Random Coil (%)
OVT	Nature/5.0	20.55 ± 0.07 ^c^	21.80 ± 0.85 ^g^	21.85 ± 0.78 ^c^	35.80 ± 0.97 ^a^
	Nature/7.0	20.35 ± 1.91 ^c^	19.85 ± 5.73 ^g^	23.20 ± 3.25 ^bc^	36.60 ± 0.57 ^a^
	Nature/9.0	19.50 ± 1.56 ^cd^	26.30 ± 1.84 ^e^	20.10 ± 2.26 ^c^	34.15 ± 1.91 ^ab^
	70 °C/5.0	8.75 ± 0.35 ^f^	53.85 ± 1.20 ^b^	8.20 ± 0.14 ^f^	29.10 ± 0.99 ^c^
	70 °C/7.0	15.4 ± 0.42 ^e^	36.80 ± 0.57 ^c^	15.35 ± 0.92 ^e^	32.45 ± 0.64 ^b^
	70 °C/9.0	13.95 ± 0.07 ^e^	37.50 ± 1.98 ^c^	14.30 ± 0.85 ^e^	34.25 ± 1.20 ^ab^
BLF	Nature /5.0	17.45 ± 2.62 ^d^	33.90 ± 2.91 ^c^	18.65 ± 1.30 ^d^	30.00 ± 0.99 ^c^
	Nature /7.0	20.05 ± 0.91 ^c^	26.35 ± 0.35 ^e^	21.70 ± 0.71 ^c^	31.90 ± 0.28 ^c^
	Nature /9.0	19.45 ± 0.35 ^cd^	29.20 ± 0.57 ^d^	21.35 ± 0.07 ^c^	29.95 ± 0.92 ^c^
	70 °C/5.0	17.95 ± 0.49 ^d^	33.50 ± 0.57 ^c^	18.75 ± 0.35 ^d^	29.85 ± 1.34 ^c^
	70 °C/7.0	6.65 ± 0.78 ^f^	61.40 ± 1.56 ^a^	6.75 ± 1.48 ^f^	25.15 ± 0.78 ^d^
	70 °C/9.0	7.80 ± 0.57 ^f^	54.85 ± 2.90 ^b^	7.90 ± 1.27 ^f^	29.45 ± 0.92 ^c^
HLF	Nature/5.0	26.35 ± 1.34 ^a^	8.60 ± 5.52 ^h^	30.35 ± 2.90 ^a^	34.70 ± 1.27 ^ab^
	Nature/7.0	28.45 ± 1.48 ^a^	4.50 ± 1.56 ^i^	32.65 ± 1.77 ^a^	34.40 ± 1.70 ^ab^
	Nature/9.0	27.05 ± 0.92 ^a^	9.45 ± 2.47 ^h^	30.55 ± 1.48 ^a^	32.90 ± 0.14 ^bc^
	70 °C/5.0	23.95 ± 0.64 ^b^	17.50 ± 4.24 ^g^	26.45 ± 1.77 ^b^	32.10 ± 1.84 ^bc^
	70 °C/7.0	26.30 ± 0.42 ^a^	10.20 ± 2.12 ^h^	30.10 ± 1.13 ^a^	33.40 ± 0.57 ^bc^
	70 °C/9.0	22.05 ± 0.21 ^b^	24.25 ± 0.64 ^f^	22.90 ± 0.42 ^c^	30.75 ± 0.07 ^c^

Note: ‘Nature’ refers to the samples without heating. Different letters on the same column indicate that the differences among the samples are significant (*p* < 0.05).

**Table 4 foods-12-00532-t004:** Turbidity of OVT, BLF and HLF before and after heated at 70 °C for 30 min at pH 5.0, 7.0 and 9.0.

Sample	Turbidity (%)					
	Nature/5.0	Nature/7.0	Nature/9.0	70 °C/5.0	70 °C/7.0	70 °C/9.0
OVT	0.34 ± 0.16 ^c^	1.71 ± 0.80 ^c^	2.04 ± 1.09 ^c^	56.72 ± 1.32 ^a^	30.98 ± 0.23 ^b^	4.83 ± 0.77 ^c^
BLF	2.28 ± 0.18 ^d^	3.06 ± 0.43 ^d^	2.61 ± 0.07 ^d^	7.85 ± 0.15 ^c^	59.53 ± 2.38 ^a^	12.80 ± 0.71 ^b^
HLF	4.28 ± 0.31 ^d^	5.27 ± 0.15 ^d^	5.92 ± 0.46 ^d^	45.76 ± 2.62 ^a^	35.66 ± 0.14 ^c^	39.87 ± 1.55 ^b^

Notes: Different letters (a, b, c, d) on the same line indicate that the differences among the samples are significant (*p* < 0.05). ‘Nature’ refers to the samples without heating.

## Data Availability

The data presented in this study are available on request from the corresponding author.

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
