# Peer review of "Providing New Insights on the Molecular Properties and Thermal Stability of Ovotransferrin and Lactoferrin"

_foods, 2023, doi:10.3390/foods12030532_

Round 1

Reviewer 1 Report

The manuscript entitled "Providing New Insights on the Molecular Properties and Thermal Stability of Ovotransferrin and Lactoferrin" is a well designed and novel idea to research and discuss. The Manuscript is written very well and comprehensibly. It would be great if the authors could go through the following comments:

- The abstract is not enough interesting and informative. It needs to be modified and please use data and number more to illustrate the comparison among OVT, BLF, and HLF.

- Please change the subtitle " SDS-PAGE analysis" to "Molecular weight distribution"

- Line 198-200: I think this is not correct at least for BLF. As it is obvious in the electrophoresis pattern, the reducing has caused a little change in the molecular weight destribution. You can see some narrow vague bonds bellow 78kDa for BLF. Could you please discuss this if you are agree?

- Line 249: This statement is wrong. "surface charge (similar amount of positively and negatively charged amino acid residues)". Because the surface charge is not necessarily neutral (Z=0). It could be positive (Z+) or negative (Z-). In these cases, the amount of positively and negatively charged amino acid residues is not equal or similar.

- I think it would be more tangible, if the authors investigate at least one bioactivity during these research to show the effect of processing on their activities. This manuscript is showing the potential of OVT to be used as a proper substitute for LF, however, it is not trustworthy until you investigate at least 1 special biological activity such as anti oxidant, antimicrobial, etc.

Author Response

Reviewer 1#

The manuscript entitled "Providing New Insights on the Molecular Properties and Thermal Stability of Ovotransferrin and Lactoferrin" is a well designed and novel idea to research and discuss. The Manuscript is written very well and comprehensibly. It would be great if the authors could go through the following comments:

- The abstract is not enough interesting and informative. It needs to be modified and please use data and number more to illustrate the comparison among OVT, BLF, and HLF.

Response: Thank you, we have included more data to illustrate the comparison.

- Please change the subtitle " SDS-PAGE analysis" to "Molecular weight distribution"

Response: Thanks for your suggestion, we have revised.

- Line 198-200: I think this is not correct at least for BLF. As it is obvious in the electrophoresis pattern, the reducing has caused a little change in the molecular weight destribution. You can see some narrow vague bonds bellow 78 kDa for BLF. Could you please discuss this if you are agree?

Response: Actually, some studies have showed that BLF has the single band in the reducing electrophoresis [1, 2]. In the reducing electrophoresis, if the sulfhydryl reducing agent could break BLF into fragments, the band around 78 kDa will obviously weaken or even disappear, but it did not happen in our study. After discussion, we give the below explanation. Since the purity of BLF used in this study is about 95%, the three vague bands bellow 78 kDa in reducing electrophoresis pattern of BLF correspond to the other proteins (e.g. BSA, IgG chains) in milk. Additionally, in the non-reducing electrophoresis pattern, there are also three bands, but much vaguer, in the BLF electrophoresis pattern, we suppose that the difference of clarity between non-reducing and reducing electrophoresis pattern for these three vague bands is due to the influence of gel dyeing and destaining.

[1]Neelam Mahala, Aastha Mittal, Manohar Lal, Uma S. Dubey, Isolation and characterization of bioactive lactoferrin from camel milk by novel pH-dependent method for large scale production, Biotechnology Reports, 36, 2022, e00765.

[2]David A. Goulding, Jonathan O'Regan, Lionel Bovetto, Nora M. O'Brien, James A. O'Mahony, Influence of thermal processing on the physicochemical properties of bovine lactoferrin, International Dairy Journal, 119, 2021, 105001.

- Line 249: This statement is wrong. "surface charge (similar amount of positively and negatively charged amino acid residues)". Because the surface charge is not necessarily neutral (Z=0). It could be positive (Z+) or negative (Z-). In these cases, the amount of positively and negatively charged amino acid residues is not equal or similar.

Response: ok, thanks for your suggestion, we have removed this statement.

- I think it would be more tangible, if the authors investigate at least one bioactivity during these research to show the effect of processing on their activities. This manuscript is showing the potential of OVT to be used as a proper substitute for LF, however, it is not trustworthy until you investigate at least 1 special biological activity such as anti oxidant, antimicrobial, etc.

Response: Thanks for your kind suggestion. Indeed, to explore the substitutability of   OVT for lactoferrin, lots of work is needed to be done. The research on the molecular properties and thermal stability of OVT, BLF and HLF is just the part of our work. Recently, we are also focusing on some research about the antioxidation, allergenicity, in vitro digestion characteristic and immunoregulatory activity of OVT, BLF and HLF, and we are planning to show the results in the following papers. We hope it would be accepted. Many thanks again.

Reviewer 2 Report

In this study, the authors compared the molecular properties and thermal stability of ovotransferrin, bovine lactoferrin and human lactoferrin. Strength of the study is that a good range of biochemical/chemical analyses were carried out. This allowed the authors to highlight the similarities of ovotransferrin to bovine lactoferrin and human lactoferrin, supporting their proposal to use the former as a substitute of the latter.

 Below are my minor feedbacks for the authors’ consideration:

ABSTRACT:

1.     Line 10: Did the author mean “supplement” or “substitute? Please recheck.

2.     Line 15: “Tm” should be introduced in full at the first mention.

3.     It would be useful to briefly state why those parameters evaluated can be used to justify whether ovotransferrin is a favorable substitute of lactoferrin.

INTRODUCTION:

1.     Many statements/much info here seem in need of cited references, e.g., please check lines 34-47.

2.     Lines 53 – 57: Any references can be cited to support these?

3.     Line 64: “Liu et al” is not the reference [15] cited. Please correct it.

4.     Line 75: Please check whether “remain” should be changed to “retain”.

5.     Lines 76 –77: Can the authors briefly explain in the text why “OVT is expected to have comparable or even better thermal stability to LF”?

M&M:

1.     Lines 91- 92: Did the authors prepared the ovotransferrin themselves? Since it is a key sample for this study, please cite a reference for the method used. And in fact, it would be appropriate to briefly describe how it was accomplished by putting the info in a separate section, e.g., “2.2. Preparation of ovotransferrin”.

2.     Line 103: What was the amount of protein loaded per well?

3.     Lines 148 - 149: Can the authors briefly indicate the rationales of using the chosen pH and ionic strengths? Are these arbitrary or chosen based on previous reports/experiments?

4.     Line 149: “degased” should be “degassed”.

RESULTS AND DISCUSSION:

1.     Total SH content was determined. But it is unclear form the text how this information is specifically relevant to the goal of using OVT as a substitute of lactoferrin. It would be useful to briefly explain it in lines 202-219. Will it have any implications on preservation of protein quality during food processing/heating? Or there are other reasons?

2.     Please see reposition all figures (except Fig 8) and Table 4. Figures/tables should not be shown in the paper BEFORE they are mentioned in the text.

3.     Lines 233 – 240: Discussion points/interpretation here should be supported by cited references. Otherwise, they would just appear as speculation/guesses.

4.     Line 253: “Previous studies…” is mentioned but no references are cited for the studies.

5.     Lines 259 – 260: “some physical treatment techniques (e.g. ultrasound, pulsed electric field) can be introduced to enhance the dispersibility of OVT during processing” – Will this then make it more costly/inconvenient to use OVT, thus not favoring using it as a substitute of lactoferrins?

6.     Lines 280 – 281: “OVT with lower H0 would be advantageous to BLF and HLF in terms of processing properties in solution system.” – This statement seems unclear/too general to me. Can the authors elaborate it a little?

7.     Table 2 – Are the data collected from triplicates? If so, why not present them as mean + std deviation as are in other tables? This also should be followed by statistical analysis to make the interpretation provided less subjective.

8.     Lines 392 – 395: “OVT and BLF were not thermally stable at acidic condition, while HLF remained thermally steady at different pH.” -  Can the authors briefly discuss implications of this on the food types/systems OVT can be applied to?

9.     Table 3 appears incomplete. Statistical analysis (ANOVA and Duncan’s multiple-range tests) should be done to make the authors’ interpretation of results in the table less subjective and well-supported.

10.  Figure 9 – Are the data shown here based one or three replicates? This info is unclear in the caption.

11.  Figures 518 – 522: “The changes on turbidity of OVT solution … near the pI (5.31) of OVT.” – Can any references be cited to support the discussion here?

12.  Line 542: “Overall, heat treatment predominates over the denaturation aggregation process of OVT, BLF and HLF.” – This statement seems unclear to me. Can it be revised to make it clearer?

CONCLUSION:

1.     Lines 554-555: “the advanced structures of these three proteins experienced obvious changes” – Can the authors be more specific what the “advanced structures” and “obvious changes” refer to?

2.     Lines 560 – 561: “further investigation about … was required” – It should be “is required”.

Author Response

Reviewer 2#

In this study, the authors compared the molecular properties and thermal stability of ovotransferrin, bovine lactoferrin and human lactoferrin. Strength of the study is that a good range of biochemical/chemical analyses were carried out. This allowed the authors to highlight the similarities of ovotransferrin to bovine lactoferrin and human lactoferrin, supporting their proposal to use the former as a substitute of the latter.

Below are my minor feedbacks for the authors’ consideration:

ABSTRACT:

  1. Line 10: Did the author mean “supplement” or “substitute? Please recheck.

Response: Thanks for your suggestion, actually, we want to explore the substitutability of   OVT for lactoferrin. We have revised the “supplement” to “substitute”.

  1. Line 15: “Tm” should be introduced in full at the first mention.

Response: ok, we have revised.

  1. It would be useful to briefly state why those parameters evaluated can be used to justify whether ovotransferrin is a favorable substitute of lactoferrin.

Response: ok, we have included one sentence in the Abstract section.

INTRODUCTION:

  1. Many statements/much info here seem in need of cited references, e.g., please check lines 34-47.

Response: ok, we have included 5 references.

  1. Lines 53 – 57: Any references can be cited to support these?

Response: Thanks for your suggestion, we have included references.

  1. Line 64: “Liu et al” is not the reference [15] cited. Please correct it.

Response: ok, we have revised.

  1. Line 75: Please check whether “remain” should be changed to “retain”.

Response: Thanks for your suggestion, we have changed.

  1. Lines 76 –77: Can the authors briefly explain in the text why “OVT is expected to have comparable or even better thermal stability to LF”?

Response: Actually, the preceding sentence “Therefore, better thermal stability is vital to retain the desirable biological activities of functional proteins.” indicates that if OVT has comparable or even better thermal stability to LF, it is most likely for OVT to retain its natural biological activities and show better product processing quality. We have included one sentence to explain it. Thank for your suggestion.

M&M:

  1. Lines 91- 92: Did the authors prepared the ovotransferrin themselves? Since it is a key sample for this study, please cite a reference for the method used. And in fact, it would be appropriate to briefly describe how it was accomplished by putting the info in a separate section, e.g., “2.2. Preparation of ovotransferrin”.

Response: ok, we have included the preparation method of ovotransferrin in section 2.2.

  1. Line 103: What was the amount of protein loaded per well?

Response: The loading volume for each well is 10 μL and the protein concentration is 2 mg/mL.

  1. Lines 148 - 149: Can the authors briefly indicate the rationales of using the chosen pH and ionic strengths? Are these arbitrary or chosen based on previous reports/experiments?

Response: we chose the conditions based on a previous study and we have included the reference.

  1. Line 149: “degased” should be “degassed”.

Response: ok, we have revised, thank you.

RESULTS AND DISCUSSION:

  1. Total SH content was determined. But it is unclear form the text how this information is specifically relevant to the goal of using OVT as a substitute of lactoferrin. It would be useful to briefly explain it in lines 202-219. Will it have any implications on preservation of protein quality during food processing/heating? Or there are other reasons?

Response: Total SH content could reflect the number of disulfide bond which is related to the structure stability of protein in different processing conditions (e.g. pH, temperature, ionic strength). We have included discussion about it.

  1. Please see reposition all figures (except Fig 8) and Table 4. Figures/tables should not be shown in the paper BEFORE they are mentioned in the text.

Response: ok, we have revised.

  1. Lines 233 – 240: Discussion points/interpretation here should be supported by cited references. Otherwise, they would just appear as speculation/guesses.

Response: ok, we have included several references to support the discussion.

  1. Line 253: “Previous studies…” is mentioned but no references are cited for the studies.

Response: ok, we have revised.

  1. Lines 259 – 260: “some physical treatment techniques (e.g. ultrasound, pulsed electric field) can be introduced to enhance the dispersibility of OVT during processing” – Will this then make it more costly/inconvenient to use OVT, thus not favoring using it as a substitute of lactoferrins?

Response: yes, thanks for your comment, we have removed this statement.

  1. Lines 280 – 281: “OVT with lower H0 would be advantageous to BLF and HLF in terms of processing properties in solution system.” – This statement seems unclear/too general to me. Can the authors elaborate it a little?

Response: ok, we have revised.

  1. Table 2 – Are the data collected from triplicates? If so, why not present them as mean + std deviation as are in other tables? This also should be followed by statistical analysis to make the interpretation provided less subjective.

Response: ok, we just provided the average value of the data and did not do statistical analysis because we focused more on the discussion of Fig.4. We have revised.

  1. Lines 392 – 395: “OVT and BLF were not thermally stable at acidic condition, while HLF remained thermally steady at different pH.” -  Can the authors briefly discuss implications of this on the food types/systems OVT can be applied to?

Response: ok, we have included a sentence to discuss about this.

  1. Table 3 appears incomplete. Statistical analysis (ANOVA and Duncan’s multiple-range tests) should be done to make the authors’ interpretation of results in the table less subjective and well-supported.

Response: ok, we have done the statistical analysis.

  1. Figure 9 – Are the data shown here based one or three replicates? This info is unclear in the caption.

Response: ok, we have included the information.

  1. Figures 518 – 522: “The changes on turbidity of OVT solution … near the pI (5.31) of OVT.” – Can any references be cited to support the discussion here?

Response: ok, we have cited a reference here.

  1. Line 542: “Overall, heat treatment predominates over the denaturation aggregation process of OVT, BLF and HLF.” – This statement seems unclear to me. Can it be revised to make it clearer?

Response: ok, we have revised.

CONCLUSION:

  1. Lines 554-555: “the advanced structures of these three proteins experienced obvious changes” – Can the authors be more specific what the “advanced structures” and “obvious changes” refer to?

Response: ok, we have revised in the text.

  1. Lines 560 – 561: “further investigation about … was required” – It should be “is required”.

Response: ok, we have revised, thank you.